# Three-Dimensional-Evaluation of Aortic Changes after Frozen Elephant Trunk (FET) in Zone 0 vs. Zone 2 in Acute Type A Aortic Dissection

**DOI:** 10.3390/jcm13092677

**Published:** 2024-05-02

**Authors:** Ahmed Ghazy, Ryan Chaban, Philipp Pfeiffer, Chris Probst, Daniel-Sebastian Dohle, Hendrik Treede, Bernhard Dorweiler

**Affiliations:** 1Department of Cardiac and Vascular Surgery, University of Mainz, Faculty of Medicine and University Hospital Mainz, 55131 Mainz, Germany; 2Department of Vascular and Endovascular Surgery, University of Cologne, Faculty of Medicine and University Hospital Cologne, 50937 Cologne, Germany

**Keywords:** aortic dissection, FET, remodeling, 3D analysis

## Abstract

**Introduction:** The management of aortic dissection has evolved significantly over the decades, with the frozen elephant trunk (FET) procedure emerging as a key technique for treating complex aortic pathologies. Recent practices involve deploying the FET prosthesis more proximally in the aorta (Zone 0) to reduce complications, leading to questions about its impact on long-term aortic remodeling compared to traditional Zone 2 deployment. **Methods:** This retrospective analysis utilized 3D segmentation software to assess the volumetric changes in aortic remodeling after acute Type A aortic dissections, comparing FET stent graft deployment in Zone 0 and Zone 2. The study included 27 patients operated on between 2020 and 2022, with volumetric measurements taken from postoperative and 6-month follow-up CT scans. Statistical analyses were performed to evaluate the differences in the aortic true lumen (TL) and the perfused false lumen (PFL) between the two groups. **Results:** Both Zone 0 and Zone 2 deployments resulted in significant true lumen (TL) increases (Z0 *p* = 0.001, Z2 *p* < 0.001) and perfused false lumen (PFL) decreases (Z0 *p* = 0.02, Z2 *p* = 0.04), with no significant differences in volumetric changes between the groups (*p* = 0.7 post op and *p* = 0.9 after 6 months). The distal anastomosis in Zone 0 did not compromise the aortic remodeling outcomes and was associated with reduced distal ischemia and cerebral perfusion times (*p* = 0.041). The angle measurements in Zone 0 did not show any significant changes after the 6-month control (*p* = 0.2). However, in Zone 2, a significant change was detected. (*p* = 0.022). The part comparison analyses did not indicate significant differences in aortic deviation between the groups (*p* = 0.62), suggesting comparable effectiveness in aortic remodeling. **Conclusions:** Performing the distal anastomosis more proximally in Zone 0 offers technical advantages without compromising the effectiveness of aortic remodeling compared to the traditional Zone 2 deployment. This finding supports the continued recommendation of Zone 0 deployment in the management of acute Type A aortic dissections, with ongoing studies being needed to confirm the long-term outcomes and survival benefits.

## 1. Introduction

Managing aortic dissection remains a challenging endeavor [1]. In the last decades, numerous treatment methodologies have been proposed and established. Most famously, the 2-stage elephant trunk approach was first outlined in 1983 [2]. This method sparked the idea of integrating a hybrid prosthesis featuring a distal stent in the descending aorta. Over time, this concept underwent significant evolution, until the first description of the frozen elephant trunk (FET) procedure in 1996 [3,4].

The FET technique changed the treatment landscape for complex aortic pathologies. Compared to the traditional 2-stage approach, or total arch replacement, the FET procedure simplified the therapeutic process significantly [1]. In recent times, the FET method has solidified its position as an invaluable tool in the management of a spectrum of aortic pathologies, particularly in the management of the acute aortic Type A dissection [4,5].

Despite this considerable advancement, FET remains a complicated surgical procedure, with a high mortality rate of 8–10% and neurological complications or spinal injury rate of 7% and 4%, respectively [6]. Its efficacy depends on the stabilization of the aorta’s true lumen and the subsequent thrombosis of the false lumen, a process known as “aortic remodeling” [7]. By the insertion of the stented portion of the hybrid prosthesis into the true lumen, reentries are sealed, and the true lumen (TL) is expanded. False lumen (FL) thrombosis at the level of the stent graft, or even beyond, leads to FL shrinkage by thrombus resorption. This process of TL growth and FL shrinkage is called positive aortic remodeling.

The largest single-center experience with the FET technique using the “E-vita Open” prosthesis (Artivion, Kennesaw, GA, USA) was reported in Essen, Germany, and showed a successful aortic remodeling in 90% of patients with acute dissection, and a correlation of 5-year and 10-year freedom from re-intervention rates of 87% and 74%, respectively [8,9]. The success of the remodeling was influenced by multiple variables, notably the deployment location of the implanted stent and allowing the volumetric increase in the distal hybrid prosthesis.

Historically, the distal anastomosis of the stent graft of the FET Prothesis was performed in aortic Zone 3, before moving proximally to be performed in Zone 2 [1,3,4]. In pursuit of surgical simplicity, and to mitigate technical challenges, many medical institutions have started deploying the FET prosthesis more proximally in Zone 0 [10,11]. In our center, we shifted from deploying the stent in Zone 2 to Zone 0 in 2021. This strategic transition manifested in tangible benefits, notably, a marked decrease in distal ischemia and cerebral perfusion durations, as well as less renal insufficiency requiring temporary dialysis. However, the following question persists: Does the more proximal deployment in Zone 0 ensure long-term aortic remodeling comparable to the traditional FET stenting in aortic Zone 2?

To better understand the changes and their effects on aortic remodeling, we performed an imaging-based retrospective analysis of aortic volumetric changes using 3D segmentation software and compared the results of deploying the FET stent graft in Zone 2 and Zone 0 following acute Type A aortic dissections.

## 2. Patients and Methods

The research protocol was approved by the local institutional medical ethics committee. The need for informed consent was waived due to the retrospective nature of the analysis and the use of anonymous data.

The inclusion criteria were as follows: (1) operation for acute Type A dissection, (2) survival for at least 6 months, and (3) complete follow-up with high-resolution CT both directly after the surgery and 6 months after. All patients who had proximal anastomosis performed in other aortic zones were excluded.

In our study, 27 patients were included (Zone 0: 12 patients, Zone 2: 15 patients) who were operated on between 2020 and 2022.

### Surgical Technique

The operation was performed with the patient in a supine position under endotracheal intubation anesthesia. Skin disinfection was carried out for more than 3 min, according to a standardized hygiene protocol. A puncture of the left femoral artery originating from the true lumen was carried out, and an insertion of a 4 French pigtail catheter via a 5 French sheath, secured by transesophageal echocardiography (TEE), into the true lumen at the level of the left subclavian artery (LSA) was performed. An incision was made on the right infraclavicular area, exposing the right axillary artery, while preserving the plexus, and performing an end-to-side anastomosis with an 8-mm prosthesis. The prosthesis was then flushed with heparin solution.

After that, systemic heparin was intravenously administrated, and then a median sternotomy was performed. The pericardial sac was opened from the diaphragm until the highest point that could be reached superiorly. Retraction stitches were placed on the edges of the pericardial sac and tightly attached to the sternal retractor or to the skin to elevate the heart anteriorly. Cautious preparation of the aortic arch with encircling of the brachiocephalic trunk, left carotid artery, and left subclavian artery, was carried out. Cannulation of the right atrium with a two-stage cannula was performed, connecting the arterial line to the 8-mm prosthesis of the right axillary artery. Initiation of extracorporeal circulation (ECC) and cooling to 26 °C was carried out. An insertion of a left ventricular vent through the superior right pulmonary vein was performed. The aorta was then clamped, and the dissected ascending aorta was incised enough to achieve a good exposure of the coronary ostia. The selective cardioplegia was administrated into both ostia, 1600 mL over >5 min, with ice-water cooling. The ascending aorta was resected up to the sinotubular junction, the hematoma was removed from the dissected right and non-coronary sinus, and the reattachment of the commissures with 4/0 Teflon-coated prolene sutures was performed. In case it was required, the root was glued with Bio-glue.

The sinotubular junction was measured and the appropriate aortic prosthesis was selected. An end-to-end anastomosis with the selected prosthesis reinforced with Teflon-coated prolene 4/0 sutures was performed. After verification of the anastomosis with cardioplegia and the achievement of the target temperature, the ligature of the left subclavian artery was performed. The left common carotid artery (LCA) was clamped, followed by transition to selective cerebral perfusion (18 °C) by clamping the brachiocephalic trunk, the opening of the arch, and resecting the brachiocephalic trunk and left carotid artery.

The advancement of the pigtail catheter and Amplatzer guide-wire was performed next. The selection of the appropriate frozen elephant trunk prosthesis and the insertion of the prosthesis into the true lumen guided with the wire was carried out. The release of the prosthesis in Zone 0 or Zone 2 was then performed. Distal anastomosis was then reinforced with Teflon-coated prolene 3/0 in the selected aortic Zone (Figure 1).

The insertion of a Foley catheter into the stent graft segment for perfusion via the adjunct pump to end the distal ischemia was performed.

Then, the anastomosis of the head–neck vessels to the arch prosthesis was performed using mostly a 12-mm prosthesis for the brachiocephalic trunk and a 6-mm prosthesis for the left carotid artery. Afterwards, de-airing in the head-down position and transition to whole-body perfusion was performed. At the end, an end-to-end anastomosis of the ascending aorta prosthesis to the aortic arch prosthesis was performed.

After releasing the aortic clamp, the heart began beating in sinus rhythm.

Then, an incision was made in the left infraclavicular area, exposing the left axillary artery while preserving the plexus, and an end-to-side anastomosis with a 7-mm prosthesis on the undersurface of the vessel was carried out in such a way that the prosthesis could be easily guided extranatomically to the ascending aorta through the 2nd intercostal space.

The 7-mm bypass to the left subclavian artery was guided to the ascending aorta through the second intercostal space, the ascending prosthesis was tangentially clamped, and the bypass was anastomosed end-to-side with prolene 5/0.

After sufficient reperfusion and rewarming, gradual weaning off cardiopulmonary bypass (CPB) was performed. Using stable hemodynamics, the heart–lung machine was decanulated and the protamine was administrated. After hemostasis, and at the end of the procedure, chest drains and temporary pacemaker leads were placed, and the sternotomy was closed using sternal wires.

## 3. Volumetric Analysis

Semiautomatic segmentation using 3mensio Vascular (Pie Medical Imaging, Netherlands) was performed for all of the CT scans. After manual identification of the true lumen (TL) of the aorta, a centerline was created beginning in the aortic root and extending into the abdominal aorta. The TL extension was automatically identified, and manual corrections were performed on the reconstructed orthogonal CT slices, as required. The resulting segmentation allowed cross-sectional area measurements to be taken at any selected point, as well as length and volume measurements of desired parts.

Areas of the TL were measured at the proximal and distal end of the stent graft and at the level of the celiac trunk (Figure 1).

Volume measurements of the TL and the PFL at the stent graft level (S1), between the stent graft and celiac trunk (S2), and the total thoracic aorta (S1+S2) were taken.

The angle between the aortic arch (measured at the most cranial point of the arch) and the distal end of the stent graft was measured on the created 3D model. All of the measurements were obtained independently by two observers (Figure 1 and Figure 2).

## 4. Part Comparison Analysis

The DICOM dataset was segmented using Mimics Medical version 25 (Mimics Innovation Suite, Materialise, Belgium). The segmentation was performed using a semi-automated (threshold) tool and then the intra-aortic blood volume was separated from the surroundings using the split mask tool (Figure 3).

The two generated 3D models of each patient (one generated from postoperative CTA and one generated from the 6-month control CTA) were imported into Materialise 3-matic (Materialise, Belgium). After the alignment of the two chronological models using the interactive translate and global registration tool, a part comparison tool was used. The part comparison tool was used to assess the deviation (distance) of the surfaces of the discharge and 6-month follow-up 3D models. The deviation between the discharge and the 6-month follow-up 3D models was calculated for S1 alone and then combined with the thoracic aorta (S1+S2). The results are shown as a color-coding of the 3D Files (Figure 4). The mean value given in the histogram was used for the statistical analysis.

## 5. Statistical Analysis

The statistical analysis was performed with GraphPad Prism 10 (GraphPad software, Boston, MA, USA). The mean and standard deviation were calculated for continuous variables in case of normality, otherwise the median and quartiles (min. and max.) were reported. The binary variables were compared using the Pearson chi-square or Fisher exact test. Continuous variables were analyzed for normality using the Shapiro–Wilk test. In the case of normal distribution, the paired *t*-test was used. Otherwise, a Mann–Whitney U test was performed.

## 6. Results

### 6.1. Patients’ Demography and Operative Data

Discharge CTA was performed at 8 ± 4 days, and a 6-month CTA was performed 181 ± 47 days postoperatively. No significant difference in patient demographics and comorbidities were noticed between the two groups (Table 1). The durations of surgery, cardiopulmonary bypass (CPB), and aortic cross-clamping were similar. Distal ischemia was significantly reduced in the Zone 0 group (25 ± 6 min for Z0 vs. 31 ± 11 min for Z2, *p* = 0.041).

In all of the patients in Zone 2, an E-vita Open prosthesis was utilized. In the Zone 0 group, one patient received a Thoraflex prosthesis (Terumo Aortic, Sunrise, FL, USA), while all of the other patients received E-vita Open neo prostheses. The mean stent length was 14 ± 2.8 cm in the Zone 0 group and 13 ±0.5 cm in the Zone 2 group. Among the Zone 0 group, four patients received an 18-cm stent, three patients received a 13-cm stent, and five patients received a 12-cm stent. However, in the Zone 2 group, all of the patients received a 13-cm stent, except for one patient, who received a 15-cm stent.

### 6.2. Postoperative Outcome

No mortality was observed within the first 30 days in either group. No significant differences were noticed between the two groups regarding their hospital stay (16 ± 7 in Zone 0 vs. 18 ± 11 days in Zone 2, *p* = 0.6) or in intensive care unit (ICU) stay (4 ± 2 days in Zone 0 vs. 6 ± 7 days in Zone 2, *p* = 0.09). Dialysis was only necessary for two patients of the Zone 2 group. The neurological status was impaired preoperatively in two patients of the Zone 0 group, and no progress was documented until discharge. (*p* = 0.1).

### 6.3. CTA Analysis

Table 2 presents the volumetric measurements obtained from both groups from the true and false lumen.

#### 6.3.1. Volumetric Analysis

A significant TL increase between discharge and 6-month follow-up was observed in both groups in S1, S2, and the total thoracic aorta (S1+S2). The mean TL volume change in S1 was 30% in the Zone 0 and Zone 2 groups (*p* = 0.69). In S2, the mean TL volume increase was 50% in the Zone 0 group and 40% in the Zone 2 group (*p* = 0.78).

Only minimally perfused false lumen (PFL) was found at the stent graft level in Segment 1 (Zone 0: 2.5 +− 8 cc Zone 2: 0.7 +− 1.3 cc) at discharge. This minimal PFL further decreased after 6 months (Zone 0: 0.5 +− 1.7 cc, Zone 2: 0 +− 0.2 cc). In the distal of the stent graft (S2), significant PFL reduction in both groups was observed.

This observation was supported by the measurements of the aortic cross-sectional TL area, which showed a significant increase at level 1 in both groups (*p* = 0.004 vs. 0.0004). The TL area increase was only significant in the Zone 2 group at level 2.

At the 6-month follow-up endpoint, the total aortic volume in both groups after 6 months was almost identical, averaging 125 cm (*p* = 0.9).

#### 6.3.2. Angle Measurement Analysis

The postoperative angle measurements and those taken at the 6-month follow-up showed the mean angle in Z-0 FET to be more stable (77° postoperatively, 72° at 6-month follow-up) compared to the Z-2 group (84° postoperatively, 74° at 6-month follow-up).

A significant difference between both groups was observed at discharge (*p* = 0.042); however, by the 6-month mark, as the curvature of the Zone 2 FET tended to flatten, no significant difference was noted (*p* = 0.551) (Figure 5).

#### 6.3.3. Part Comparison Analysis (PCA)

The PCA revealed an average deviation of 4.4 mm across S1 and S2 segments in Z-0 FET, compared to an average deviation of 5.3 mm in Z-2 FET. Specifically, for the S1 segment, a deviation of 1.3 mm was noted in Z-0 FET, while a 2.0-mm deviation was observed in Z-2 FET. The statistical analysis between the two groups did not show a significant difference (*p* = 0.3).

## 7. Discussion

In our study, we observed an increase in the true lumen (TL) across both Segment 1 and Segment 2, even within our short (6 months) follow-up. Following the frozen elephant trunk (FET) procedure, the minimally perfused false lumen (FL) at the stent graft level (S1) showed further reduction, and even this minimal partial false lumen (PFL) disappeared after 6 months. The FET’s efficacy in reducing PFL was significant in both groups, regardless of whether it was implanted in Zone 0 (Z0) or Zone 2 (Z2).

On the stent geometry observation, and despite concerns that proximal anastomosis in Zone 0 might lead to a greater deviation of the stent graft, neither our part comparison analysis nor the angle measurements provide any evidence to support those concerns [12].

The use of the FET procedure for the management of acute aortic dissection is increasing, due to its established efficacy [12,13]. Our findings demonstrate that both Zone 0 and Zone 2 FET procedures yield favorable remodeling outcomes [13]. Notably, Zone 0 showed less deviation in outcomes, adding to the advantage of the avoidance of extensive periaortic tissue dissection and aortic arch wall resection, presenting fewer technical challenges and possibly reducing surgery duration [14,15,16,17,18,19].

To our knowledge, this is the first study to demonstrate similar behavior in TL increase and PFL decrease between Zone 0 and Zone 2 FET. It is also important to highlight the significance of our novel 3D-based methodology in assessing postoperative changes in aortic morphology. This approach provides precise and comprehensive evaluations, allowing for detailed comparisons between the different zones of distal anastomosis. By leveraging the analytical capabilities of 3D-engineering software, we were able to detect subtle variations in aortic remodeling patterns, contributing to a deeper understanding of the effects of FET procedures [20,21,22,23].

The results of this study suggest that similar outcomes in terms of aortic remodeling can be expected regardless of the zone of distal anastomosis used. This finding holds significant implications for surgical decision making, as it indicates that Zone 0 FET procedures offer comparable benefits in promoting distal remodeling.

## 8. Conclusions

The current results reveal no difference between patients operated on in Zone 0 and those operated on in Zone 2 regarding the key markers for aortic remodeling. Taking into account the technical advantage provided by moving the procedure more proximally, this method can continue to be recommended. The final results concerning aortic remodeling, a survival benefit, and long-term outcomes are still pending.

## 9. Limitations

This study is based on data from a single center with a small sample size and has a retrospective design for a limited follow-up of 6 months. The results of our study might be limited by a survival bias, due to only including patients with an available CT scan at 6 months postoperatively.

## Figures and Tables

**Figure 1 jcm-13-02677-f001:**
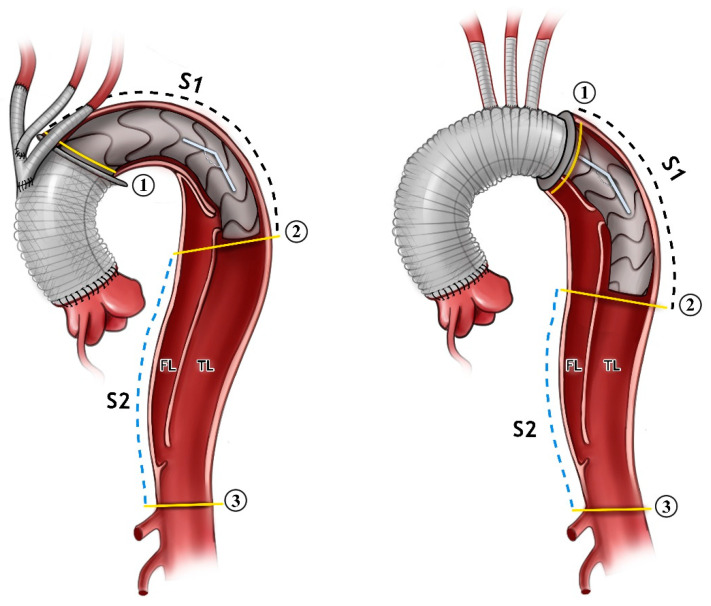
An illustration showing the FET inserted in Zone 0 to the (**left**) and Zone 2 to the (**right**). The angle between the aortic arch and desc. aorta is demonstrated. The measurements were carried out just for the stent part alone (Segment 1, S1), and distally from the stent down to the coeliac trunk (Segment 2, S2). All of the measurements were carried out for the false lumen (FL) and true lumen (TL) in both segments. The aortic cross-sectional area is shown in the yellow lines in position 1, 2, and 3.

**Figure 2 jcm-13-02677-f002:**
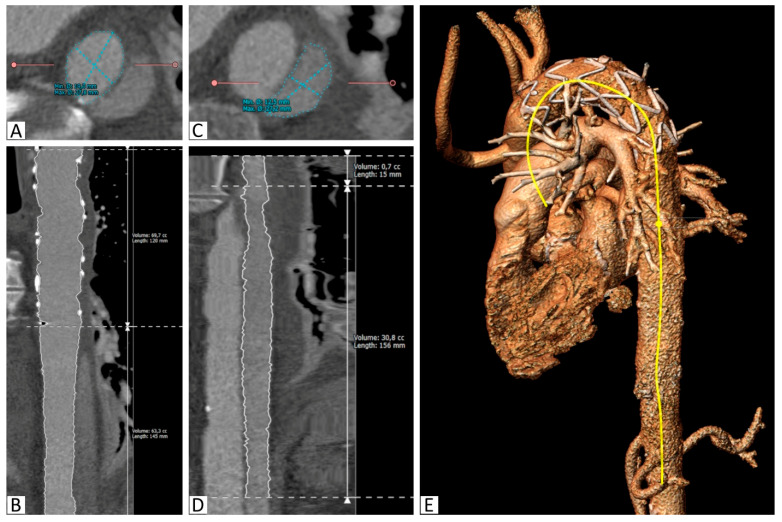
Segmentation of the aorta with the implanted stent. (**A**): Segmentation of the true lumen in the descending thoracic aorta distal to the stent graft on a reconstructed slice orthogonal to the vessel, (**B**): Curved multiplanar reconstruction of the true lumen between the proximal stent graft and the celiac trunk, including length and volume calculations, (**C**): Segmentation of the false lumen in the descending thoracic aorta distal to the stent graft on a reconstructed slice orthogonal to the vessel, (**D**): Curved multiplanar reconstruction of the false lumen between its origin in the stent graft and the celiac trunk, including length and volume calculations, (**E**): 3D reconstruction with the semiautomatically created center-line (yellow).

**Figure 3 jcm-13-02677-f003:**
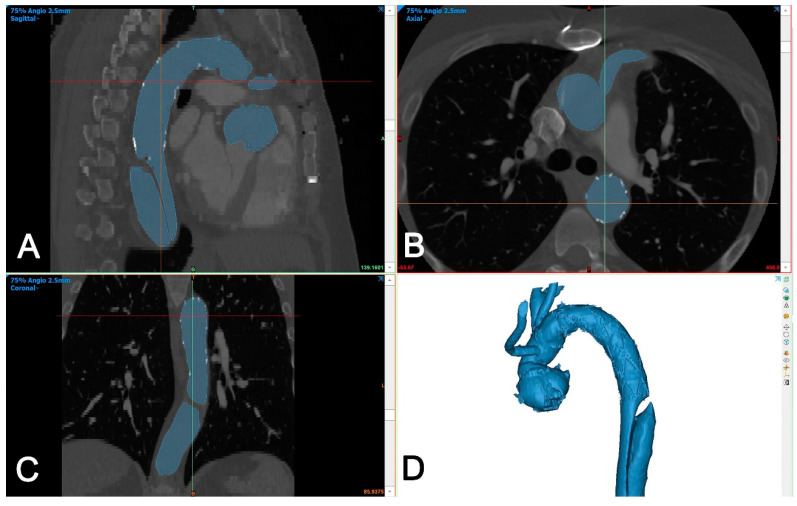
Segmentation of aortic volume using dedicated 3D-engineering software (Mimis Innovation suite, Materialise, Belgium). (**A**–**C**): Aortic volume is semi-automatically segmented (blue color) and manually corrected in three perpendicular planes (sagittal (**A**), axial (**B**), and coronal (**C**)), if needed. (**D**): A 3D-volume is generated automatically and is used for further processing.

**Figure 4 jcm-13-02677-f004:**
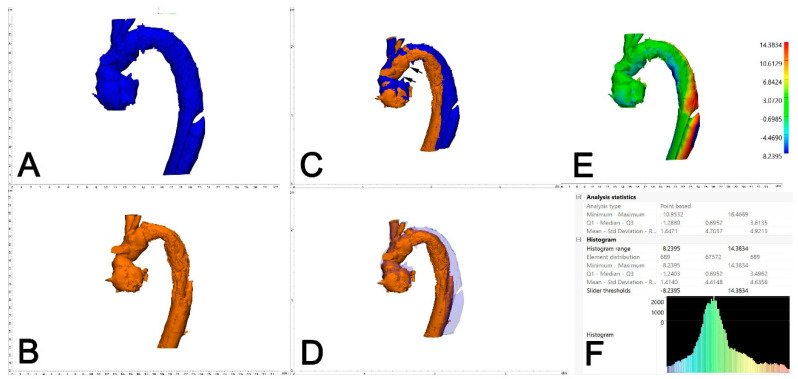
Assessment of aortic deviation between 0 and 6 months following FET. A dedicated 3D-engineering software-tool (“Part comparison analysis”) was employed. Therefore, the 3D aortic volumes at 6 months (**A**, blue) and 0 months (**B**, orange) were aligned/overlayed using the distal and the proximal anastomosis of the Dacron part of the FET (**C**, black arrows). With reduced transparency of the aortic volume at 6 months (blue), adequate alignment between the two aortic volumes is demonstrated (**D**). The result of the part comparison tool is depicted as a color-coded “heat-map” of the aortic volume (deviation in mm) (**E**). The area of maximum conformational change (deviation) is located at aortic Zones 3 and 4. For statistical analysis, the respective values (mean deviation) were retrieved from the PCA analysis statistics (**F**).

**Figure 5 jcm-13-02677-f005:**
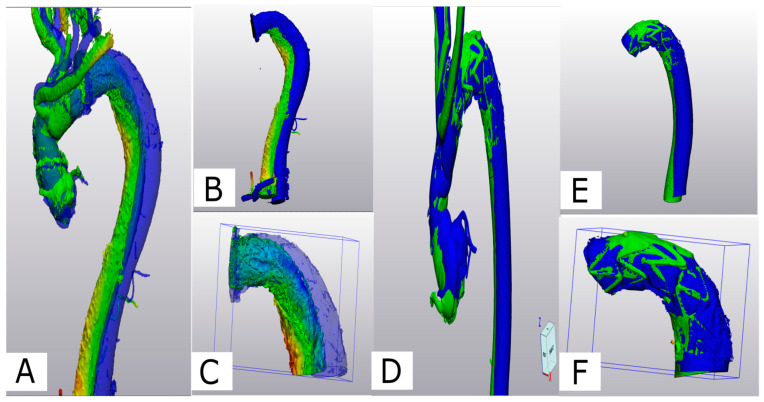
(**A**) Comparison of aortic conformational change between Z2- (**A**–**C**) and Z0-FET (**D**–**F**): Z-2-FET a deviation analysis is shown for the post-operative 3D volume in green and the 6-month 3D volume in blue. (**B**) Z-2-FET stent with desc. aorta (S1+2) showing the deviation. (**C**) Z-2-FET stent segment (S1) deviation. (**D**) Z-0-FET on the middle deviation analysis between the post-operative model in green and the 6-month control model in blue. (**E**) Z-0-FET stent with desc. aorta (S1+2) showing minimal deviation. (**F**) Z-0-FET stent segment (S1) with minimal deviation.

**Table 1 jcm-13-02677-t001:** Demographic and operative data.

	Zone 0	Zone 2	*p*-Value
**Demographic**			
*n*	12	15	
Gender (m)	8	13	0.2
Age	62 ± 9	61 ± 9	0.7
Hypertension	10 (83%)	9 (60%)	0.2
Diabetes mellitus	2 (16%)	0 (0%)	>0.9
Smoking history	3 (25%)	1 (6%)	>0.9
Hyperlipoproteinemia	2 (16%)	1 (6%)	>0.9
Coronary artery disease	1 (8%)	2 (13%)	>0.9
Renal insufficiency	0 (0%)	1 (6%)	>0.9
Body mass index (kg/m^2^)	25.5 ± 2.3	27.3 ± 4.8	0.174
Body surface area (m^2^)	2.0 ± 0.3	2.1 ± 0.2	0.376
**Operative Data**			
Operative time (Min.)	415 ± 55	425 ± 40	0.715
Cardiopulmonary bypass time (Min.)	261 ± 17	252 ± 41	0.142
Aortic cross-clamp time (Min.)	164 ± 34	162 ± 45	0.994
Cerebral perfusion time (Min.)	48 ± 8	55 ± 14	0.271
Distal ischemia time (Min.)	25 ± 6	31 ± 11	0.041

Different demographic details of our cohort and the operative data in minutes.

**Table 2 jcm-13-02677-t002:** Outcome parameters.

	Zone 0			Zone 2		
	0 Months	6 Months	PZ00/6	0 Months	6 Months	P Z2 0/6
**Aortic volume**						
**Total**	129 ± 46	142 ± 43	0.07	122 ± 35	143.5 ± 31	0.004
**TL**	101 ± 49	125 ± 47	0.001	95 ± 29	125 ± 35	0.00005
**FL**	16.05 [00–102]	4.45 [00–73]	0.02	18.1 [00–115]	13.8 [00–67]	0.04
**Aortic Segments**						
**S1 Total**	50 ± 19	60 ± 16	0.002	52 ± 13	64.5 ± 11	0.00005
**S1 TL**	47 ± 17	59.5 ± 15.7	0.00005	51.3 ± 13	64.4 ± 11	0.00003
**S1 FL**	0 [00–29]	0 [00–6]	0.3	0 [00–4]	0 [00–1]	0.07
**S2 total**	79 ± 38	82 ± 43	0.56	70 ± 27	79 ± 24	0.089
**S2 TL**	53 ± 40	66 ± 46	0.02	43.8 ± 20	60 ± 27	0.00089
**S2 FL**	16.05 [00–74]	4.45 [00–67]	0.04	18.1 [00–112]	13.8 [00–67]	0.047
**Aortic cross-sectional area (mm^3^)**						
**1**	438 ± 117	493 ± 124	0.00410678	457 ± 97	525 ± 120	0.0004404
**2**	388 ± 172	457 ± 204	0.11541371	390 ± 116	515 ± 118	0.00053499
**3**	297 ± 175	304 ± 178	0.59229521	289 ± 130	317 ± 155	0.25562499
**Stent Curvature (Degrees)**	77.7 ± 6	71.5 ± 14	0.2	84 ± 8	74.5 ± 12	0.022

Different volumetric measurements of aorta. TL: true lumen, FL: false lumen, Different segments of the aorta S1: stent segment, S2: from the distal end of the stent until the coeliac trunk. All of the mean ± standard deviation (for the stent part) volumetric measurements are in cubic centimeters (cc) of the true lumen and false lumen in Zone 0 and Zone 2, respectively.

## Data Availability

The data presented in this study are available on request from the corresponding author.

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
