# Peer review of "Three-Dimensional-Evaluation of Aortic Changes after Frozen Elephant Trunk (FET) in Zone 0 vs. Zone 2 in Acute Type A Aortic Dissection"

_jcm, 2024, doi:10.3390/jcm13092677_

Round 1
Reviewer 1 Report
Comments and Suggestions for Authors
I am grateful to the editor for the opportunity to review the manuscript “3D-Evaluation of aortic changes after frozen elephant trunk (FET) in Zone 0 vs. Zone 2 in acute Type A aortic dissection” by Ghazy Ahmed et al. Indeed, the frozen elephant trunk technique is finding more and more adherents among surgeons (ref. 1-2, see below). Unlike published works, the peer-reviewed article compares changes in the aorta during frozen elephant trunk implantation in different zones of the aorta using 3D-Evaluation. This determines the relevance and novelty of the study.
However, the presented manuscript reveals omissions to which we would like to draw the attention of the authors of the article.
1. The text of the manuscript does not contain a title, information about the authors and information about their affiliations.
2. There is no list of references, which does not allow us to assess the relevance and relevance of the references used.
3. The Discussion section requires stylistic editing. This section should begin with the main results obtained by the authors. Also, you should not refer to tables in this section - this is for the Results section.
4. A section on the limitations of the study should be added.
5. Judging by the Statistical analysis section, the distribution of some quantitative data differs from normal. In this case, the data presentation format should be different (instead of Mean and standard deviation, use the format - Me [upper and lower quartiles])
References:
1. Lee KFL, Bhatia I, Chan TLD, Au WKT, Ho KLC. Proximalization of Frozen Elephant Trunk Procedure: Zone 0 or 1 versus Zone 2 or 3 Arch Repair. Thorac Cardiovasc Surg. 2024 Mar;72(2):89-95. doi: 10.1055/s-0042-1757631.
2. Pichlmaier M, Tsilimparis N, Hagl C, Peterss S. New anatomical frozen elephant trunk graft for zone 0: endovascular technology reduces invasiveness of open surgery to the max. Eur J Cardiothorac Surg. 2022 Jan 24;61(2):490-492. doi: 10.1093/ejcts/ezab394.
Comments on the Quality of English Language
No comments
Author Response
|
1. Summary |
|
|
|
Thank you very much for taking the time to review this manuscript. Please find the detailed responses below and the corresponding revisions/corrections highlighted/in track changes in the re-submitted files. |
||
|
2. Point-by-point response to Comments and Suggestions for Authors |
||
|
Reviewer1: I am grateful to the editor for the opportunity to review the manuscript “3D-Evaluation of aortic changes after frozen elephant trunk (FET) in Zone 0 vs. Zone 2 in acute Type A aortic dissection” by Ghazy Ahmed et al. Indeed, the frozen elephant trunk technique is finding more and more adherents among surgeons (ref. 1-2, see below). Unlike published works, the peer-reviewed article compares changes in the aorta during frozen elephant trunk implantation in different zones of the aorta using 3D-Evaluation. This determines the relevance and novelty of the study. Thank you very much for your nice and encouraging wording. Thank you for providing us with two additional references, which we included in the revised manuscript.
However, the presented manuscript reveals omissions to which we would like to draw the attention of the authors of the article. 1. The text of the manuscript does not contain a title, information about the authors and information about their affiliations. I think this has to do with the version that You have received as a reviewer (an anonymous version). We submitted a standard version (with title, author list and affiliation, etc.) as per journal guidelines. 2. There is no list of references, which does not allow us to assess the relevance and relevance of the references used. I am sorry to hear that. Our submitted version contained a list of references as usual. We will check with journal to provide you with it.
3. The Discussion section requires stylistic editing. This section should begin with the main results obtained by the authors. Also, you should not refer to tables in this section - this is for the Results section. Thank you for this constructive comment. We revised the “discussion” section as requested. 4. A section on the limitations of the study should be added. Thank you for this comment. The limitations were discussed in the second section following “conclusion”. It is now in a dedicated section (line 300), as requested. 5. Judging by the Statistical analysis section, the distribution of some quantitative data differs from normal. In this case, the data presentation format should be different (instead of Mean and standard deviation, use the format - Me [upper and lower quartiles]) Thank you for this construction comment. We revised the statistic as requested. Data that don’t follow normal distribution (The False Lumen: total, S1 and S2, etc.) are now expressed as Me[up-lo quartiles], |
||
Reviewer 2 Report
Comments and Suggestions for Authors
Thank you for the opportunity to review your manuscript. It contributes valuable data to the field of cardiovascular surgery, suggesting that Zone 0 FET deployment can be a safe and effective alternative to Zone 2 deployment for acute Type A aortic dissections. Highlighting the implications of these findings and a more nuanced discussion on the clinical significance of the findings would be valuable. Specifically, elaborating on how these findings impact current surgical practices and patient management strategies would be beneficial.
The manuscript mentions the statistical tests used but lacks detailed explanation or rationale for the choice of these tests, especially in the context of the data distribution. Additional details or justification could strengthen the methodology section.
The conclusion that Zone 0 deployment does not compromise aortic remodeling outcomes and offers technical advantages is well supported by the data.
Author Response
- Summary
Thank you very much for taking the time to review this manuscript. Please find the detailed responses below and the corresponding revisions/corrections highlighted/in track changes in the re-submitted files.
|
Reviewer 2: Thank you for the opportunity to review your manuscript. It contributes valuable data to the field of cardiovascular surgery, suggesting that Zone 0 FET deployment can be a safe and effective alternative to Zone 2 deployment for acute Type A aortic dissections. Highlighting the implications of these findings and a more nuanced discussion on the clinical significance of the findings would be valuable. Specifically, elaborating on how these findings impact current surgical practices and patient management strategies would be beneficial. Thank you for your comment. We are glad you liked our manuscript.
The manuscript mentions the statistical tests used but lacks detailed explanation or rationale for the choice of these tests, especially in the context of the data distribution. Additional details or justification could strengthen the methodology section. Thank you very much. We revised the statistical section. We know provide more data about the distribution and the rational of the tests.
The conclusion that Zone 0 deployment does not compromise aortic remodeling outcomes and offers technical advantages is well supported by the data. We agree with this comment. |
Reviewer 3 Report
Comments and Suggestions for Authors
I want to state that I read the article with interest. Aortic dissection, which constitutes the most troublesome patient group in emergency departments, frequently brings together cardiovascular surgeons and emergency physicians. Although the number of cases in the article is small, I think it will lead to future prospective studies. My advice to writers;
1. They should omit the historical development in the Abstract section because if it is not omitted, it will be necessary to add a reference. However, references cannot be added to the summary section. It should also be strengthened by adding the numbers of the results and the actual p-value in the summary section.
2. Patients' Comorbid conditions and acceptance and rejection criteria should be added.
3. Since the number of cases is less than 30, you cannot use a student t-test, you should use a nonparametric test. It would be best if you revised the statistics again.
Author Response
- Summary
Thank you very much for taking the time to review this manuscript. Please find the detailed responses below and the corresponding revisions/corrections highlighted/in track changes in the re-submitted files.
|
Reviewer 3: I want to state that I read the article with interest. Aortic dissection, which constitutes the most troublesome patient group in emergency departments, frequently brings together cardiovascular surgeons and emergency physicians. Although the number of cases in the article is small, I think it will lead to future prospective studies. Thank you very much for your encouragement. My advice to writers; 1. They should omit the historical development in the Abstract section because if it is not omitted, it will be necessary to add a reference. However, references cannot be added to the summary section. It should also be strengthened by adding the numbers of the results and the actual p-value in the summary section. Thanks. We revised the abstract as requested. 2. Patients' Comorbid conditions and acceptance and rejection criteria should be added. Done, patients comorbidities (Table 1), acceptance and rejection critieria (“Patients and methods”, line 87) are now added. 3. Since the number of cases is less than 30, you cannot use a student t-test, you should use a nonparametric test. It would be best if you revised the statistics again. Done. We revised the statistics and used nonparametric test instead. |
|
|
Round 2
Reviewer 1 Report
Comments and Suggestions for Authors
The authors answered my questions and comments and made adjustments to the text. However, I was not satisfied with the authors' response to my 3rd comment. No stylistic correction has been made to the text of the Discussion section.
Comments on the Quality of English Language
No comments
Author Response
Thanks for highlighten your suggestion for another time ; I have edited the discussion again. I started the discussion with a focus on the main findings in our study without referring to the tables as suggested.
Reviewer 3 Report
Comments and Suggestions for Authors
Dear Auhors,
It seems that the authors have made the desired changes.
Author Response
Thanks , we are glad that we made the desired changes from your side.